# On a Resolution of Another Isolated Case of a Kummer's Quadratic Transformation for $_2F_1$

**Mohamed Jalel Atia** [1,2,*] and **Ahmed Saleh Al-Mohaimeed** [1]

1    Department of Mathematics, College of Science, Qassim University, Buraidah 51452, Saudi Arabia
2    Laboratory LR17ES11, University of Gabes, Gabès 6072, Tunisia
*    Correspondence: m.attia@qu.edu.sa

**Abstract:** It is well-known that the Kummer quadratic transformation formula is valid provided that its parameters fulfill some specific conditions (see Gradshteyn, Ryzhik, Tables of Integrals, Series and Products, 9.130, 9.134.1). Very recently, one of us established a new identity when one of these conditions is not fulfilled. In this paper, we aim to discuss another isolated case which completely different from the first. Moreover, in the end, we mention two interesting consequences of these two new results.

**Keywords:** hypergeometric functions; Kummer's Quadratic Transformation; hypergeometric series with finitely many terms; hypergeometric series with infinitely many terms; differential equation

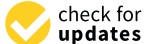



## 1. Introduction

Gauss [1] defined his famous infinite series as follows

$$1 + \frac{ab}{c}\frac{z}{1!} + \frac{a(a+1)b(b+1)}{c(c+1)}\frac{z^2}{1!} + \cdots \tag{1}$$

This infinite series (1) is usually denoted by the notation $_2F_1(a,b;c;z)$ or simply $F$ and is commonly known as the Gauss's function or the hypergeometric series. Gauss's function or the hypergeometric series is a solution of a second order differential equation. The convergence conditions of $_2F_1$ are as follows,

- a hypergeometric series terminates if $a$ or $b$ is equal to a negative integer or zero. For $c = -n(n = 0, 1, 2, \cdots)$, the hypergeometric series is indeterminate if neither $a$ nor $b$ is equal to $-m$ (where $m < n$ and $m$ is a natural number),
- if we exclude these values of the parameters $a, b, c$, a hypergeometric series converges in the unit circle $|z| < 1$. $_2F_1$ then has a branch point at $z = 1$. Then we have the following conditions for convergence on the unit circle:

1.    $0 \leq \Re(a + b - c) < 1$, the series converges throughout the entire unit circle, except at the point $z = 1$,
2.    $\Re(a + b - c) < 0$, the series converges (absolutely) throughout the entire unit circle,
3.    $\Re(a + b - c) \geq 1$, the series diverges on the entire unit circle.

It is well-known that the Kummer quadratic transformation formula

$$_2F_1(\alpha, \beta; 2\alpha; z) = (1 - \frac{z}{2})^{-\beta} \, _2F_1\left(\frac{\beta}{2}, \frac{\beta+1}{2}; \alpha + \frac{1}{2}; \left(\frac{z}{2-z}\right)^2\right). \tag{2}$$

is valid provided that $\{2\alpha + 1, \ \alpha + \frac{3}{2}\}$ are not natural numbers and $\alpha - \beta$ is not an integer (see Gradshteyn, Ryzhik, Tables of Integrals, Series and Products, 9.130, 9.134.1). Very recently, one of us established a new identity for an isolated case where $\alpha - \beta$ is an integer

by letting $\alpha$ to be a negative integer and $\beta$ to be an even positive integer which we extended here into any even integer not necessarily positive and where we gave explicitly the expressions $u_\alpha^{(\beta)}(z)$ such that [2]

$$_2F_1(\alpha, \beta; 2\alpha; z) = (1 - \frac{z}{2})^{-\beta} \, _2F_1\left(\frac{\beta}{2}, \frac{\beta+1}{2}; \alpha + \frac{1}{2}; (\frac{z}{2-z})^2\right) + u_\alpha^{(\beta)}(z).$$

In this paper, we aim to discuss another isolated case with $\alpha$ to be a negative integer and $\beta$ to be an odd integer where we gave explicitly the expressions $v_\alpha^{(\beta)}(z)$ such that

$$_2F_1(\alpha, \beta; 2\alpha; z) = (1 - \frac{z}{2})^{-\beta} \, _2F_1\left(\frac{\beta}{2}, \frac{\beta+1}{2}; \alpha + \frac{1}{2}; (\frac{z}{2-z})^2\right) + v_\alpha^{(\beta)}(z).$$

**The results for $u_\alpha^{(\beta)}(z)$ and for $v_\alpha^{(\beta)}(z)$ are completely different. The expressions for $u_\alpha^{(\beta)}(z)$ and $v_\alpha^{(\beta)}(z)$ are given in the paper.** Moreover, in the end, we mention two interesting consequences of our main result involving a symmetric role for $u_\alpha^{(\beta)}(z)$ with an odd $\beta$ and $v_\alpha^{(\beta)}(z)$ with an even $\beta$.

In this paper, we deal with Gauss's function and exactly with its connection with the following interesting and useful Kummer's quadratic transformation [2] for the hypergeometric function $_2F_1$ (2). This quadratic transformation is valid for $\{2\alpha + 1, \ \alpha + \frac{3}{2}\}$ are not natural numbers and $\alpha - \beta$ **is NOT an integer**. In this paper, we consider the case when $\alpha - \beta$ **is an integer**.

This transformation formula is recorded in several standard texts on special functions and handbooks of mathematics, for example, in the standard text of I.S. Gradshteyn and I.M. Ryzhik [3] (9.134 and 9.134.1) and G. Andrews and al. [4] (3.1.7 page 127 with a slight modification), in the handbook by Abramowitz-Stegun [5] (15.3.20) and in DLMF: NIST Digital Library of Mathematical Functions, https://dlmf.nist.gov/ [6], accessed on 15 December 2022, 15.8.13.

As usual, $\mathbb{N}$ is the set of all natural number $\{1, 2, 3, \cdots\}$, $\mathbb{Z}$ the set of all integer $\{\cdots, -2, -1, 0, 1, 2, \cdots\}$ and $\mathbb{Z}_-$ the set of all integer $\{\cdots, -2, -1, 0\}$. In a previous paper we considered the case where $\alpha - \beta$ is an integer by taking $\beta$ an even integer (i.e., $\beta \in 2\mathbb{Z}$) and $\alpha$ is a negative integer and we gave the right identity [7]. In this contribution, we consider the case where $\alpha - \beta$ is an integer by taking $\beta \in 1 + 2\mathbb{Z}$ (i.e., $\beta$ is an odd integer) and $\alpha$ is a negative integer and we give and prove the new identity. The aim of this paper is to prove the following theorem.

**Theorem 1.** *For the isolated cases ($\alpha$ is a negative integer and $\beta \in 2\mathbb{Z}$ or $\beta \in 1 + 2\mathbb{Z}$), the Kummer's hypergeometric quadratic transformations become*

$$_2F_1\left(\frac{\beta}{2}, \frac{\beta+1}{2}; \alpha + \frac{1}{2}; (\frac{z}{2-z})^2\right) = (1 - \frac{z}{2})^\beta \, _2F_1(\alpha, \beta; 2\alpha; z)$$

$$+ \frac{2\sqrt{\pi}(\frac{\beta}{2})_{-\alpha+1}}{\Gamma(-\alpha + \frac{1}{2})} \frac{(4z - 4)^{\alpha-\beta}(2-z)^{\beta+1}}{z^{2\alpha+1-\beta}} \, _2F_1\left(1 - \frac{\beta}{2}, \frac{1}{2} + \alpha - \frac{\beta}{2}; \frac{3}{2}; (\frac{2}{z} - 1)^2\right), \quad (3)$$

$\alpha \in \mathbb{Z}^-$, $\beta \in 2\mathbb{Z}$, *which we wrote under the form*

$$_2F_1\left(\frac{\beta}{2}, \frac{\beta+1}{2}; \alpha + \frac{1}{2}; (\frac{z}{2-z})^2\right) = (1 - \frac{z}{2})^\beta \, _2F_1(\alpha, \beta; 2\alpha; z) + u_\alpha^{(\beta, even)}, \ \alpha \in \mathbb{Z}^-, \ \beta \in 2\mathbb{Z},$$

*and*

$$_2F_1\left(\frac{\beta}{2},\frac{\beta+1}{2};\alpha+\frac{1}{2};(\frac{z}{2-z})^2\right) = (1-\frac{z}{2})^\beta\, _2F_1(\alpha,\beta;2\alpha;z)$$
$$-\frac{\sqrt{\pi}(\frac{\beta+1}{2})_{-\alpha}}{\Gamma(-\alpha+\frac{1}{2})}\frac{(4z-4)^{\alpha-\beta}(2-z)^\beta}{z^{2\alpha-\beta}}\, _2F_1\left(\frac{1}{2}-\frac{\beta}{2},\alpha-\frac{\beta}{2};\frac{1}{2};(\frac{2}{z}-1)^2\right),\qquad(4)$$

$\alpha\in\mathbb{Z}^-$, $\beta\in 1+2\mathbb{Z}$, *which we wrote under the form*

$$_2F_1\left(\frac{\beta}{2},\frac{\beta+1}{2};\alpha+\frac{1}{2};(\frac{z}{2-z})^2\right) = (1-\frac{z}{2})^\beta\, _2F_1(\alpha,\beta;2\alpha;z)+v_\alpha^{(\beta,odd)},\ \alpha\in\mathbb{Z}^-,\ \beta\in 2\mathbb{Z},$$

*and as a consequence we prove that*

$$_2F_1\left(\frac{\beta}{2},\frac{\beta+1}{2};\alpha+\frac{1}{2};(\frac{z}{2-z})^2\right) = -v_\alpha^{(\beta,even)},\ \alpha\in\mathbb{Z}^-,\ \beta\in 2\mathbb{Z},$$

*and*

$$_2F_1\left(\frac{\beta}{2},\frac{\beta+1}{2};\alpha+\frac{1}{2};(\frac{z}{2-z})^2\right) = u_\alpha^{(\beta,odd)},\ \alpha\in\mathbb{Z}^-,\ \beta\in 1+2\mathbb{Z}.$$

Please note here that when $\beta$ is either an even or odd non-positive integer then the series $_2F_1(\frac{\beta}{2},\frac{\beta+1}{2};\alpha+\frac{1}{2};(\frac{z}{2-z})^2)$ is well-defined with finitely many terms and the series $_2F_1\left(\frac{1}{2}-\frac{\beta}{2},\alpha-\frac{\beta}{2};\frac{1}{2};(\frac{2}{z}-1)^2\right)$ is a series with infinitely many terms and when $\beta$ is either an even or odd positive integer then the series $_2F_1\left(\frac{\beta}{2},\frac{\beta+1}{2};\alpha+\frac{1}{2};(\frac{z}{2-z})^2\right)$ is well-defined with infinitely many terms and the series $_2F_1\left(\frac{1}{2}-\frac{\beta}{2},\alpha-\frac{\beta}{2};\frac{1}{2};(\frac{2}{z}-1)^2\right)$ is a series with finitely many terms thus the restriction due to the their convergence should be satisfied once, i.e., either $|\frac{z}{2-z}|<1$ or $|\frac{2-z}{z}|<1$ (not simultaneously).

Let us first give the Maple instructions in order to assure the readers that the results we are giving are true and right.

```
restart; F1 := proc (alpha, beta, z) options operator, arrow;
hypergeom([(1/2)*beta, (1/2)*beta+1/2], [alpha+1/2], z^2/(2-z)^2)
end proc; F2 := proc (alpha, beta, z) options operator, arrow;
(1-(1/2)*z)^beta*hypergeom([alpha, beta], [2*alpha], z) end proc; u
:= proc (alpha, beta, z) options operator, arrow;
2*(z/(2-z))^(beta-1-2*alpha)*sqrt(Pi)*pochhammer((1/2)*beta,
-alpha+1)
*hypergeom([1-(1/2)*beta, 1/2+alpha-(1/2)*beta], [3/2], (2-z)^2/z^2)
*((4*z-4)/(2-z)^2)^(alpha-beta)/GAMMA(-alpha+1/2) end proc;
v := proc (alpha, beta, z) options operator, arrow;
-(z/(2-z))^(beta-2*alpha)*sqrt(Pi)*pochhammer((1/2)*beta+1/2,
-alpha)
*hypergeom([alpha-(1/2)*beta, 1/2-(1/2)*beta], [1/2], (2-z)^2/z^2)
*((4*z-4)/(2-z)^2)^(alpha-beta)/GAMMA(-alpha+1/2) end proc;
simplify(F1(-8, 4, z)-F2(-8, 4, z)-u(-8, 4, z));
0
simplify(F1(-8, -5, z)-F2(-8, -5, z)-v(-8, -5, z));
0
simplify(F1(-8, 4, z)+v(-8, 4, z));
0
simplify(F1(-8, -5, z)+u(-8, -5, z));
0
```

We change the notations to be much more convenient. The letter $n$ denotes, in general, the integers so let us denote by $\alpha := -n + 1$, $n$ is an integer greater than 0. The expression $\alpha - \beta$ should be an integer then we take $\beta = 2a$ with

$$a \in \left\{ \cdots, -\frac{5}{2}, -2, -\frac{3}{2}, -1, -\frac{1}{2}, 0, \frac{1}{2}, 1, \frac{3}{2}, 2, \frac{5}{2}, \cdots \right\}$$

Taking into account the quantity $\left(\frac{z}{2-z}\right)^2$, if we replace $\pm\dfrac{z}{2-z}$ by $z$ (2) becomes

$$\,_2F_1\left(a, a + \frac{1}{2}; -n + \frac{3}{2}; z^2\right) = \frac{1}{(1 \pm z)^{2a}} \,_2F_1\left(2a, -n + 1; -2n + 2; \frac{\pm 2z}{1 \pm z}\right). \tag{5}$$

Here we have two situations for the integer $2a$:

- either $a$ is itself an integer, i.e., $a \in \mathbb{Z}$ and in this case we have

  1. either $a$ is itself a positive integer, i.e., $a \in \mathbb{N}$, this situation was considered in a previous paper [7] where we proved that (5) remains true for $n = 0$ but for $n = 1$ (5) becomes

$$
\begin{aligned}
\,_2F_1\left(a, a + \frac{1}{2}; \frac{1}{2}; z^2\right) &= \frac{(1+z)^{-2a} + (1-z)^{-2a}}{2} \\
&= \frac{1}{(1 \mp z)^{2a}} \pm \frac{2x(2a-1)\Gamma(a+1) \,_2F_1\left(1-a, -a+\frac{1}{2}; \frac{3}{2}; \frac{1}{z^2}\right)}{\Gamma(a)(z^2-1)^{2a}} \\
&= \frac{1}{(1 \mp z)^{2a}} \pm \frac{(z-1)^{-2a} - (z+1)^{-2a}}{2},
\end{aligned}
$$

and for $n \geq 2$ (5) should be written as

$$
\begin{aligned}
\,_2F_1\left(a, a + \frac{1}{2}; -n + \frac{3}{2}; z^2\right) &= \frac{1}{(1 \pm z)^{2a}} \,_2F_1\left(2a, -n + 1; -2n + 2; \frac{\pm 2z}{1 \pm z}\right) \\
&\pm \frac{2\sqrt{\pi}\,\Gamma(n+a) z^{2n+2a-3} \,_2F_1\left(1-a, \frac{3}{2} - n - a; \frac{3}{2}; \frac{1}{z^2}\right)}{\Gamma(a)\Gamma\left(n - \frac{1}{2}\right)(z^2-1)^{n+2a-1}},
\end{aligned} \tag{6}
$$

which, using the notation

$$u_n^{(a)} := \frac{2\sqrt{\pi}\,\Gamma(n+a) z^{2n+2a-3} \,_2F_1\left(1-a, \frac{3}{2} - n - a; \frac{3}{2}; \frac{1}{z^2}\right)}{\Gamma(a)\Gamma\left(n - \frac{1}{2}\right)(z^2-1)^{n+2a-1}} \tag{7}$$

we wrote it under the form

$$\,_2F_1\left(a, a + \frac{1}{2}; -n + \frac{3}{2}; z^2\right) = \frac{1}{(1 \pm z)^{2a}} \,_2F_1\left(2a, -n + 1; -2n + 2; \frac{\pm 2z}{1 \pm z}\right) \pm u_n^{(a)}, \tag{8}$$

  2. or $a$ is itself a negative integer, i.e., $a \in \mathbb{Z}_-$ and this situation is solved only by using the Pochhammer symbol in the formula of $u_n^{(a)}$:

$$u_n^{(a)} := \frac{2\sqrt{\pi}\,(a)_n z^{2n+2a-3} \,_2F_1\left(1-a, \frac{3}{2} - n - a; \frac{3}{2}; \frac{1}{z^2}\right)}{\Gamma\left(n - \frac{1}{2}\right)(z^2-1)^{n+2a-1}}, \tag{9}$$

and (8) remains true,

- or $a$ is not an integer but half of an integer, i.e., $a \in \{\cdots, \frac{7}{2}, -\frac{5}{2}, -\frac{3}{2}, -\frac{1}{2}, \frac{1}{2}, \frac{3}{2}, \frac{5}{2}, \frac{7}{2}, \cdots\}$ and, in this case, we prove that (5) remains true for $n = 0$ but for $n = 1$ and using the fact that $(-1)^{2a} = -1$ (5) becomes

$$_2F_1(a, a + \frac{1}{2}; \frac{1}{2}; z^2) = \frac{(1+z)^{-2a} + (1-z)^{-2a}}{2} = \frac{1}{(1 \mp z)^{2a}} \pm \frac{(1+z)^{-2a} + (z-1)^{-2a}}{2},$$

and for $n \geq 2$, $a \in \{\cdots, -\frac{5}{2}, -\frac{3}{2}, -\frac{1}{2}, \frac{1}{2}, \frac{3}{2}, \frac{5}{2}, \frac{7}{2}, \cdots\}$ and $z \neq 0$ we prove that

$$_2F_1(a, a + \frac{1}{2}; -n + \frac{3}{2}; z^2) = \frac{1}{(1 \pm z)^{2a}} \,_2F_1(2a, -n + 1; -2n + 2; \frac{\pm 2z}{1 \pm z})$$
$$\mp \frac{z^{2a-2+2n}\sqrt{\pi}(a + \frac{1}{2})_{n-1} \,_2F_1(\frac{1}{2} - a, 1 - n - a; \frac{1}{2}; \frac{1}{z^2})(z^2 - 1)^{-n+1-2a}}{\Gamma(n - \frac{1}{2})}, \tag{10}$$

which using the notation

$$v_n(a) = -\frac{z^{2a-2+2n}\sqrt{\pi}(a + \frac{1}{2})_{n-1} \,_2F_1(\frac{1}{2} - a, 1 - n - a; \frac{1}{2}; \frac{1}{z^2})(z^2 - 1)^{-n+1-2a}}{\Gamma(n - \frac{1}{2})}, \tag{11}$$

we write

$$_2F_1(a, a + \frac{1}{2}; -n + \frac{3}{2}; z^2) = \frac{1}{(1 \pm z)^{2a}} \,_2F_1(2a, -n + 1; -2n + 2; \frac{\pm 2z}{1 \pm z}) \pm v_n(a), \tag{12}$$

As an interesting consequence of our new expressions $u_n^{(a)}$ and $v_n^{(a)}$ we prove that

$$_2F_1(a, a + \frac{1}{2}; -n + \frac{3}{2}; z^2) = -u_n^{(a)}, \quad a \in \{\cdots, -\frac{5}{2}, -\frac{3}{2}, -\frac{1}{2}, \frac{1}{2}, \frac{3}{2}, \frac{5}{2}, \cdots\}$$
$$_2F_1(a, a + \frac{1}{2}; -n + \frac{3}{2}; z^2) = -v_n^{(a)}, \quad a \in \mathbb{Z}. \tag{13}$$

Please note the following that this paper is a continuation of [7].

Many authors dealt with the quadratic transformation (2) recorded in [8–11] but always with the restrictions $\{2\alpha + 1, \alpha + \frac{3}{2}\}$ are not natural numbers and $\alpha - \beta$ is not an integer (see Gradshteyn, Ryzhik, 9.130). This paper deals with some isolated cases related to $\alpha - \beta$ as an integer and will be organized as follows. First we prove promptly that the result published in [7] does not hold for $a \in \{\cdots, \frac{7}{2}, -\frac{5}{2}, -\frac{3}{2}, -\frac{1}{2}, \frac{1}{2}, \frac{3}{2}, \frac{5}{2}, \frac{7}{2}, \cdots\}$. Second, for any $a \in \mathbb{R}$ we give and prove some relations involving $_2F_1(a, a + \frac{1}{2}; -n + \frac{3}{2}; z^2)$ and $\frac{1}{(1 \pm z)^{2a}} \,_2F_1(2a, -n + 1; -2n + 2; \frac{\pm 2z}{1 \pm z})$. Finally we prove (12) and (13).

## 2. No Concordance with Previous Result

For $a \in \{\frac{1}{2}, \frac{3}{2}, \frac{5}{2}, \frac{7}{2}, \cdots\}$ let us prove first that

$$_2F_1(a, a + \frac{1}{2}; -n + \frac{3}{2}; z^2) \neq \frac{1}{(1 \pm z)^{2a}} \,_2F_1(2a, -n + 1; -2n + 2; \frac{\pm 2z}{1 \pm z}),$$

and

$$_2F_1(a, a + \frac{1}{2}; -n + \frac{3}{2}; z^2) \quad \neq \frac{1}{(1 \pm z)^{2a}} \,_2F_1(2a, -n + 1; -2n + 2; \frac{\pm 2z}{1 \pm z})$$
$$\pm \frac{2\sqrt{\pi}\Gamma(n + a)z^{2n+2a-3} \,_2F_1(1 - a, \frac{3}{2} - n - a; \frac{3}{2}; \frac{1}{z^2})}{\Gamma(a)\Gamma(n - \frac{1}{2})(z^2 - 1)^{n+2a-1}}.$$

**Proof.** In fact, with $n = 2$ and $a = \frac{5}{2}$ and taking into account the "+" sign we get

$$_2F_1(\frac{5}{2}, 3; -\frac{1}{2}; z^2) - \frac{1}{(1+z)^5} {}_2F_1(5, -1; -2; \frac{2z}{1+z}) = -\frac{2(3z^4 + 42z^2 + 35)z^3}{(z^2 - 1)^6}.$$

and

$$_2F_1(\frac{5}{2}, 3; \frac{1}{2}; z^2) - \frac{1}{(1+z)^5} {}_2F_1(5, 1; -2; \frac{2z}{1+z}) - u_2^{(\frac{5}{2})} = -\frac{(6z - 1)}{(z - 1)^6}.$$

One can use the following Maple instructions

```
restart; F1 := proc (n, a, z) options operator, arrow;
hypergeom([a+1/2, a], [-n+3/2], z^2) end proc; F2 := proc (n, a, z)
options operator, arrow; hypergeom([2*a, -n+1], [-2*n+2],
2*z/(1+z))/(1+z)^(2*a) end proc; u := proc (n, a, z) options
operator, arrow; 2*z^(2*a-3+2*n)*sqrt(Pi)*pochhammer(a, n)
*hypergeom([1-a, 3/2-n-a], [3/2], 1/z^2)
*(z^2-1)^(-n+1-2*a)/GAMMA(n-1/2) end proc;
factor(simplify(F1(2, 5/2, z)-F2(2, 5/2, z))); factor(simplify(F1(2,
5/2, z)-F2(2, 5/2, z)-u(2, 5/2, z)));
```

$\square$

Here is another curious counter-example. If we take $n = 2$, $a = \frac{1}{2}$ and $z = -\frac{1}{2}$ we find

$$_2F_1(\frac{1}{2}, 1; -\frac{3}{2}; \frac{1}{4}) = \frac{4}{9},$$

$$\frac{1}{(1 - \frac{1}{2})} {}_2F_1(1, -1; -2; \frac{-1}{1 - \frac{1}{2}}) = 0$$

$$and \ for \ z = -\frac{1}{2}, u_2^{(\frac{1}{2})} = -\frac{4}{9}.$$

## 3. Relations between These Hypergeometric Series

In order to use all the results of [7], we need to find some relations involving the first sum $_2F_1(a, a + \frac{1}{2}; -n + \frac{3}{2}; z^2)$ and the second sum $\frac{1}{(1\pm z)^{2a}} {}_2F_1(2a, -n + 1; -2n + 2; \frac{\pm 2z}{1\pm z})$.

### 3.1. Relations Involving the First Sum

In the following lemma, we give three relations involving $_2F_1(a, a + \frac{1}{2}; -n + \frac{3}{2}; z^2)$.

**Lemma 1.** *For any positive integer $n$ and for any $a \in \mathbb{R}$ we have the following results*

$$\frac{d}{dz}\left(_2F_1(a, a + \frac{1}{2}; -n + \frac{3}{2}; z^2)\right) = -\frac{2a(2a+1)z}{(2n-3)} {}_2F_1(a + 1, a + \frac{3}{2}; -n + \frac{5}{2}; z^2), \quad (14)$$

$$_2F_1(a + \frac{1}{2}, a + 1; -n + \frac{3}{2}; z^2) = {}_2F_1(a, a + \frac{1}{2}; -n + \frac{3}{2}; z^2)$$
$$-\frac{(2a+1)z^2}{(2n-3)} {}_2F_1(a + 1, a + \frac{3}{2}; -n + \frac{5}{2}; z^2), \quad (15)$$

$$_2F_1(a, a + \frac{1}{2}; -(n + 1) + \frac{3}{2}; z^2) = {}_2F_1(a, a + \frac{1}{2}; -n + \frac{3}{2}; z^2)$$
$$+\frac{(2a)(2a+1)z^2}{(2n-1)(2n-3)} {}_2F_1(a + 1, a + \frac{3}{2}; -n + \frac{5}{2}; z^2). \quad (16)$$

**Proof.** The proof of (14) is a direct consequence of the following formula [6], 15.5.1, [3–5]

$$\frac{d}{dz}\left({}_2F_1(a,b;c;z)\right) = \frac{ab}{c}\,{}_2F_1(a+1,b+1;c+1;z).\tag{17}$$

Second, let us prove (15).

$${}_2F_1\left(a,a+\frac{1}{2};-n+\frac{3}{2};z^2\right) - \frac{2a+1}{2n-3}z^2\,{}_2F_1\left(a+1,a+\frac{3}{2};-n+\frac{5}{2};z^2\right)$$

$$= \sum_{k\geq 0}\frac{(a)_k(a+\frac{1}{2})_k}{(-n+\frac{3}{2})_k}\frac{z^{2k}}{k!} - \frac{2a+1}{2n-3}\sum_{k\geq 0}\frac{(a+1)_k(a+\frac{3}{2})_k}{(-n+\frac{5}{2})_k}\frac{z^{2k+2}}{k!}$$

$$= \sum_{k\geq 0}\frac{(a)_k(a+\frac{1}{2})_k}{(-n+\frac{3}{2})_k}\frac{z^{2k}}{k!} + \sum_{k\geq 1}\frac{a+\frac{1}{2}}{-n+\frac{3}{2}}\frac{(a+1)_{k-1}(a+\frac{3}{2})_{k-1}}{(-n+\frac{5}{2})_{k-1}}\frac{z^{2k}}{(k-1)!}$$

using $(z)_k = z(z+1)_{k-1}$ we get

$${}_2F_1\left(a,a+\frac{1}{2};-n+\frac{3}{2};z^2\right) - \frac{2a+1}{2n-3}z^2\,{}_2F_1\left(a+\frac{1}{2},a+1;-n+\frac{1}{2};z^2\right)$$

$$= \sum_{k\geq 0}\frac{(a)_k(a+\frac{1}{2})_k}{(-n+\frac{3}{2})_k}\frac{z^{2k}}{k!} + \sum_{k\geq 1}\frac{k(a+1)_{k-1}(a+\frac{1}{2})_k}{(-n+\frac{3}{2})_k}\frac{z^{2k}}{k!}$$

$$= 1 + \sum_{k\geq 1}\left((a)_k + k(a+1)_{k-1}\right)\frac{(a+\frac{1}{2})_k}{(-n+\frac{3}{2})_k}\frac{z^{2k}}{k!}$$

using $(z)_k + k(z+1)_{k-1} = (z+1)_k$ we get

$${}_2F_1\left(a,a+\frac{1}{2};-n+\frac{3}{2};z^2\right) - \frac{2a+1}{2n-3}z^2\,{}_2F_1\left(a+\frac{1}{2},a+1;-n+\frac{1}{2};z^2\right)$$

$$= 1 + \sum_{k\geq 1}(a+1)_k\frac{(a+\frac{1}{2})_k}{(-n+\frac{3}{2})_k}\frac{z^{2k}}{k!}$$

$$= \sum_{k\geq 0}(a+1)_k\frac{(a+\frac{1}{2})_k}{(-n+\frac{3}{2})_k}\frac{z^{2k}}{k!} = {}_2F_1\left(a+\frac{1}{2},a+1;-n+\frac{3}{2};z^2\right).$$

Let us prove (16). It is easy to see that

$${}_2F_1\left(a,a+\frac{1}{2};-(n+1)+\frac{3}{2};z^2\right) - {}_2F_1\left(a,a+\frac{1}{2};-n+\frac{3}{2};z^2\right)$$

$$= \frac{(2a)(2a+1)z^2}{(2n-1)(2n-3)}\,{}_2F_1\left((a+1),(a+1)+\frac{1}{2};-(n-1)+\frac{3}{2};z^2\right).$$

In fact

$${}_2F_1\left(a,a+\frac{1}{2};-(n+1)+\frac{3}{2};z^2\right) - {}_2F_1\left(a,a+\frac{1}{2};-n+\frac{3}{2};z^2\right)$$

$$= \sum_{k\geq 1}\frac{(a)_k(a+\frac{1}{2})_k z^{2k}}{k!}\left(\frac{1}{(-n+\frac{1}{2})_k} - \frac{1}{(-n+\frac{3}{2})_k}\right)$$

$$= \sum_{k\geq 1}\frac{(a)_k(a+\frac{1}{2})_k z^{2k}}{(-n+\frac{3}{2})_{k-1}k!}\left(\frac{1}{-n+\frac{1}{2}} - \frac{1}{-n+k+\frac{1}{2}}\right)$$

$$= \sum_{k\geq 1}\frac{(a)_k(a+\frac{1}{2})_k z^{2k}}{(-n+\frac{3}{2})_{k-1}(k-1)!}\frac{1}{(-n+\frac{1}{2})(-n+k+\frac{1}{2})}$$

$$= \sum_{k\geq 1}\frac{a(a+\frac{1}{2})(a+1)_{k-1}(a+\frac{5}{2})_{k-1}z^{2k}}{(-n+\frac{1}{2})(-n+\frac{3}{2})(-n+\frac{5}{2})_{k-1}(k-1)!}.$$

□

*3.2. Relations Involving the Second Sum*

In the following lemma we give three relations involving $\frac{1}{(1\pm z)^{2a+1}} \, _2F_1\left(2a+1, -n+1; -2n+2; \frac{\pm 2z}{1\pm z}\right)$.

**Lemma 2.** *For any positive integer n and for any $a \in \mathbb{R}$ we have the following results*

$$\frac{d}{dz}\left(\frac{1}{(1\pm z)^{2a}} \, _2F_1(2a, -n+1; -2n+2; \frac{\pm 2z}{1\pm z})\right)$$
$$= -\frac{2a(2a+1)z}{(2n-3)} \frac{1}{(1\pm z)^{2a+2}} \, _2F_1\left(2a+2, -n+2; -2n+4; \frac{\pm 2z}{1\pm z}\right), \qquad (18)$$

$$\frac{1}{(1\pm z)^{2a+1}} \, _2F_1\left(2a+1, -n+1; -2n+2; \frac{\pm 2z}{1\pm z}\right)$$
$$= \frac{1}{(1\pm z)^{2a}} \, _2F_1\left(2a, -n+1; -2n+2; \frac{\pm 2z}{1\pm z}\right) \qquad (19)$$
$$- \frac{2a+1}{2n-3} \frac{z^2}{(1\pm z)^{2a+2}} \, _2F_1\left(2a+2, -n+2; -2n+4; \frac{\pm 2z}{1\pm z}\right),$$

$$\frac{1}{(1\pm z)^{2a}} \, _2F_1\left(2a, -(n+1)+1; -2(n+1)+2; \frac{\pm 2z}{1\pm z}\right)$$
$$= \frac{1}{(1\pm z)^{2a}} \, _2F_1\left(2a, -n+1; -2n+2; \frac{\pm 2z}{1\pm z}\right) \qquad (20)$$
$$+ \frac{(2a)(2a+1)}{(2a-1)(2n-3)} \frac{z^2}{(1\pm z)^{2a+2}} \, _2F_1\left(2(a+1), -(n-1)+1; -2(n-1)+2; \frac{\pm 2z}{1\pm z}\right).$$

**Proof.** First, let us prove (18) and let us prove it for the "+"sign,

$$\frac{d}{dz}\left(\frac{1}{(1+z)^{2a+1}} \, _2F_1(2a+1, -n+1; -2n+2; \frac{2z}{1+z})\right)$$
$$= \frac{2a}{(1+z)^{2a+2}}\left(\, _2F_1(2a+1, -n+2; -2n+3; \frac{2z}{1+z})\right)$$
$$- (z+1) \, _2F_1\left(2a, -n+1; -2n+2; \frac{2z}{1+z}\right),$$

To prove (19) we begin by considering the following change of variable for the $+$ sign

$$\frac{1}{y} = \frac{2z}{z+1}, \qquad (21)$$

whereas for the $-$ sign we assume

$$\frac{1}{y} = \frac{-2z}{-z+1}.$$

For the + sign, we should prove that

$$
{}_2F_1\left(2a, -n+1; -2n+2; \frac{1}{y}\right) - \frac{(2a+1)}{4(2n-3)y^2}\, {}_2F_1\left(2a+2, -n+2; -2n+4; \frac{1}{y}\right)
$$

$$
- \frac{(2y-1)}{2y}\, {}_2F_1\left(2a+1, -n+1; -2n+2; \frac{1}{y}\right) = 0. \tag{22}
$$

Let us prove (22). The LHS of (22) gives

$$
{}_2F_1\left(2a, -n+1; -2n+2; \frac{1}{y}\right) - \frac{(2a+1)}{4(2n-3)y^2}\, {}_2F_1\left(2a+2, -n+2; -2n+4; \frac{1}{y}\right)
$$

$$
- {}_2F_1\left(2a+1, -n+1; -2n+2; \frac{1}{y}\right) + \frac{1}{2y}\, {}_2F_1\left(2a+1, -n+1; -2n+2; \frac{1}{y}\right).
$$

Using the following relation

$$
{}_2F_1\left(2a, -n+1; -2n+2; \frac{1}{y}\right) - {}_2F_1\left(2a+1, -n+1; -2n+2; \frac{1}{y}\right)
$$

$$
= -\frac{1}{2y}\, {}_2F_1\left(2a+1, -n+2; -2n+3; \frac{1}{y}\right)
$$

the LHS of (22) becomes

$$
- \frac{(2a+1)}{4(2n-3)y^2}\, {}_2F_1\left(2a+2, -n+2; -2n+4; \frac{1}{y}\right)
$$

$$
+ \frac{1}{2y}\, {}_2F_1\left(2a+1, -n+1; -2n+2; \frac{1}{y}\right) - \frac{1}{2y}\, {}_2F_1\left(2a+1, -n+2; -2n+3; \frac{1}{y}\right).
$$

Using the following relation

$$
\frac{1}{2y}\, {}_2F_1\left(2a+1, -n+1; -2n+2; \frac{1}{y}\right) - \frac{1}{2y}\, {}_2F_1\left(2a+1, -n+2; -2n+3; \frac{1}{y}\right)
$$

$$
= \frac{(2a+1)}{4(2n-3)y^2}\, {}_2F_1\left(2a+2, -n+2; -2n+4; \frac{1}{y}\right)
$$

the LHS vanishes.　□

**Lemma 3.** *In the following lemma, we give these relations involving $u_n^{(a)}$ and $v_n^{(a)}$.*

$$
u_{n+1}^{(a)} = u_n^{(a)} + \frac{(2a)(2a+1)z^2}{(2n-1)(2n-3)} u_{n-1}^{(a+1)}. \tag{23}
$$

*as well as*

$$
v_{n+1}^{(a)} = v_n^{(a)} + \frac{(2a)(2a+1)z^2}{(2n-1)(2n-3)} v_{n-1}^{(a+1)}. \tag{24}
$$

**Proof.** The proof of (23) is a direct consequence of the combination of (8) with (16) and (20), whereas the proof of (24) is a direct consequence of the combination of (12) with (16) and (20).　□

*3.3. Relations between New Added Terms*

Using this lemma, (2) and results given in [7] we give the following result

**Theorem 2.** *For any $a \in \mathbb{R}$ we have the following results*

$$
v_n^{(a+\frac{1}{2})} = u_n^{(a)} - \frac{2a+1}{2n-3} z^2 u_{n-1}^{(a+1)}
$$

*as well as*

$$u_n^{(a+\frac{1}{2})} = v_n^{(a)} - \frac{2a+1}{2n-3}z^2 v_{n-1}^{(a+1)}.$$

**Proof.** Using (19) and (2) we have

$$_2F_1\left(a+\frac{1}{2}, a+1; -n+\frac{3}{2}; z^2\right) = \; _2F_1\left(a, a+\frac{1}{2}; -n+\frac{3}{2}; z^2\right) - \frac{2a+1}{2n-3}z^2 \; _2F_1\left(a+1, a+\frac{3}{2}; -n+\frac{5}{2}; z^2\right)$$

and

$$\frac{1}{(1 \pm z)^{2a+1}} \; _2F_1\left(2a+1, -n+1; -2n+2; \frac{\pm 2z}{1 \pm z}\right) = \frac{1}{(1 \pm z)^{2a}} \; _2F_1\left(2a, -n+1; -2n+2; \frac{\pm 2z}{1 \pm z}\right)$$
$$- \frac{2a+1}{2n-3}z^2 \frac{1}{(1 \pm z)^{2a+2}} \; _2F_1\left(2a+2, -n+2; -2n+4; \frac{\pm 2z}{1 \pm z}\right).$$

Using results of [7] we have

$$_2F_1\left(a+\frac{1}{2}, a+1; -0+\frac{3}{2}; z^2\right) = \frac{1}{(1 \pm z)^{2a}} \; _2F_1\left(2a, -0+1; -0+2; \frac{\pm 2z}{1 \pm z}\right),$$

$$_2F_1\left(a+\frac{1}{2}, a+1; -1+\frac{3}{2}; z^2\right) = \frac{1}{(1 \pm z)^{2a}} \; _2F_1\left(2a, -1+1; -2+2; \frac{\pm 2z}{1 \pm z}\right) - u_1^{(a)},$$

and for $n \geq 2$ we have

$$_2F_1\left(a+\frac{1}{2}, a+1; -n+\frac{3}{2}; z^2\right) = \frac{1}{(1 \pm z)^{2a}} \; _2F_1\left(2a, -n+1; -2n+2; \frac{\pm 2z}{1 \pm z}\right) + u_n^{(a)}.$$

If we combine all these quantities together we obtain

$$v_n^{(a+\frac{1}{2})} = u_n^{(a)} - \frac{2a+1}{2n-3}z^2 u_{n-1}^{(a+1)}.$$

□

In the following proposition, we give the simplified expression of $v_n^{(a)}$

**Lemma 4.**

$$v_n^{(a)} = -\frac{\sqrt{\pi}\left(a+\frac{1}{2}\right)_{n-1} z^{2n+2a-2} \; _2F_1\left(\frac{1}{2}-a, -n-a+1; \frac{1}{2}; \frac{1}{z^2}\right)}{\Gamma\left(n-\frac{1}{2}\right)(z^2-1)^{n+2a-1}}. \tag{25}$$

**Proof.** We have just proved that

$$v_n^{(a+\frac{1}{2})} = u_n^{(a)} - \frac{(2a+1)}{(2n-3)}z^2 u_{n-1}^{(a+1)}.$$

Using (7) we get

$$v_n^{(a)} = \frac{2\sqrt{\pi}(a)_n z^{2n+2a-3} \, {}_2F_1(1-a, \frac{3}{2}-n-a; \frac{3}{2}; \frac{1}{z^2})}{\Gamma(n-\frac{1}{2})(z^2-1)^{n+2a-1}}$$

$$-\frac{(2a+1)}{(2n-3)} z^2 \frac{2\sqrt{\pi}(a+1)_{n-1} z^{2n+2a-3} \, {}_2F_1(-a, \frac{3}{2}-n-a; \frac{3}{2}; \frac{1}{z^2})}{\Gamma(n-\frac{3}{2})(z^2-1)^{n+2a}}$$

$$= \frac{2\sqrt{\pi}(a)_n z^{2n+2a-3}}{\Gamma(n-\frac{1}{2})(z^2-1)^{n+2a}} \left( (z^2-1) \, {}_2F_1(1-a, \frac{3}{2}-n-a; \frac{3}{2}; \frac{1}{z^2}) - \right.$$

$$\left. \frac{(a+\frac{1}{2})z^2}{a} \, {}_2F_1(-a, \frac{3}{2}-n-a; \frac{3}{2}; \frac{1}{z^2}) \right)$$

$$= \frac{2\sqrt{\pi}(a)_n z^{2n+2a-3}}{\Gamma(n-\frac{1}{2})(z^2-1)^{n+2a}} \left( z^2 \, {}_2F_1(1-a, \frac{3}{2}-n-a; \frac{3}{2}; \frac{1}{z^2}) - z^2 \, {}_2F_1(-a, \frac{3}{2}-n-a; \frac{3}{2}; \frac{1}{z^2}) - \right.$$

$$\left. {}_2F_1(1-a, \frac{3}{2}-n-a; \frac{3}{2}; \frac{1}{z^2}) - \frac{z^2}{2a} \, {}_2F_1(-a, \frac{3}{2}-n-a; \frac{3}{2}; \frac{1}{z^2}) \right)$$

$$= \frac{2\sqrt{\pi}(a)_n z^{2n+2a-3}}{\Gamma(n-\frac{1}{2})(z^2-1)^{n+2a}} \left( \frac{2a+2n-3}{3} \, {}_2F_1(1-a, \frac{5}{2}-n-a; \frac{5}{2}; \frac{1}{z^2}) - {}_2F_1(1-a, \frac{3}{2}-n-a; \frac{3}{2}; \frac{1}{z^2}) \right.$$

$$\left. -\frac{z^2}{2a} \, {}_2F_1(-a, \frac{3}{2}-n-a; \frac{3}{2}; \frac{1}{z^2}) \right)$$

$$= \frac{2\sqrt{\pi}(a)_n z^{2n+2a-3}}{\Gamma(n-\frac{1}{2})(z^2-1)^{n+2a}} \left( -\frac{2(n+a)}{3} \, {}_2F_1(1-a, \frac{3}{2}-n-a; \frac{5}{2}; \frac{1}{z^2}) - \frac{z^2}{2a} \, {}_2F_1(-a, \frac{3}{2}-n-a; \frac{3}{2}; \frac{1}{z^2}) \right)$$

$$= \frac{\sqrt{\pi}(a)_n z^{2n+2a-1}}{a\Gamma(n-\frac{1}{2})(z^2-1)^{n+2a}} \left( -\frac{4a(n+a)}{3z^2} \, {}_2F_1(1-a, \frac{3}{2}-n-a; \frac{5}{2}; \frac{1}{z^2}) - {}_2F_1(-a, \frac{3}{2}-n-a; \frac{3}{2}; \frac{1}{z^2}) \right)$$

$$= \frac{\sqrt{\pi}(a+1)_{n-1} z^{2n+2a-1}}{\Gamma(n-\frac{1}{2})(z^2-1)^{n+2a}} \left( {}_2F_1(-a, \frac{1}{2}-n-a; \frac{1}{2}; \frac{1}{z^2}) \right) = v_n^{(a+\frac{1}{2})}$$

where

$$v_n^{(a)} = -\frac{\sqrt{\pi}(a+\frac{1}{2})_{n-1} z^{2n+2a-2} \, {}_2F_1(\frac{1}{2}-a, -n-a+1; \frac{1}{2}; \frac{1}{z^2})}{\Gamma(n-\frac{1}{2})(z^2-1)^{n+2a-1}}.$$

$\square$

**Notation 1.** *In the sequel, we denote by*

$$(Fv)_n^{(a)} = \, {}_2F_1(a, a+\frac{1}{2}; -n+\frac{3}{2}; z^2) + v_n^{(a)}, \ a \in \mathbb{Z}$$

*and*

$$(Fu)_n^{(a)} = \, {}_2F_1(a, a+\frac{1}{2}; -n+\frac{3}{2}; z^2) + u_n^{(a)}, \ a \in \{\cdots, -\frac{5}{2}, -\frac{3}{2}, -\frac{1}{2}, \frac{1}{2}, \frac{3}{2}, \frac{5}{2}, \cdots\}.$$

**Consequence 1.** *For any $a \in \mathbb{Z}$ we have the following result*

$$(Fv)_n^{(a)} = \, {}_2F_1(a, a+\frac{1}{2}; -n+\frac{3}{2}; z^2) + v_n^{(a)} = 0.$$

*For any $a \in \{\cdots, -\frac{5}{2}, -\frac{3}{2}, -\frac{1}{2}, \frac{1}{2}, \frac{3}{2}, \frac{5}{2}, \cdots\}$ we have the following result*

$$(Fu)_n^{(a)} = \, {}_2F_1(a, a+\frac{1}{2}; -n+\frac{3}{2}; z^2) + u_n^{(a)} = 0.$$

**Proof.** Let us begin by proving that $(Fv)_n^{(a)} = 0$, $a \in \mathbb{Z}$ and $n \in \mathbb{N} \cup \{0\}$. Let us compute first $(Fv)_0^{(a)}$ and $(Fv)_1^{(a)}$:

$$(Fv)_0^{(a)} = {}_2F_1\left(a, a + \frac{1}{2}; \frac{3}{2}; z^2\right) + v_0^{(a)} = \frac{1}{2z(-1 + 2a)}\left(-(1 + z)^{1-2a} + (1 - z)^{1-2a}\right)$$

$$+ \frac{1}{2z(-1 + 2a)}(z^2 - 1)^{-2a}\left((z - 1)(1 + z)^{2a} + (1 + z)(1 - z)^{2a}\right) = 0$$

because $2a - 1$ is an odd integer (among the conditions of the convergence of the series we should add $z \notin \{-1, 0, 1\}$).

The same calculations lead to

$$(Fv)_1^{(a)} = {}_2F_1\left(a, a + \frac{1}{2}; \frac{1}{2}; z^2\right) + v_1^{(a)} = \frac{1}{2}\left((1 + z)^{-2a} + (1 - z)^{-2a}\right)$$

$$- \frac{(1 + z)^{2a} + (-1 + z)^{2a}}{2}\left((1 + z)(-1 + z)\right)^{-2a} = 0$$

because $2a$ is an even integer.

Using (24) and (16) we easily get for all $n \in \mathbb{N}$ and $a \in \mathbb{Z}$

$$(Fv)_{n+1}^{(a)} = (Fv)_n^{(a)} + \frac{(2a)(2a + 1)z^2}{(2n - 1)(2n - 3)}(Fv)_{n-1}^{(a+1)}. \tag{26}$$

With $n = 1$ in (26) we get

$$(Fv)_2^{(a)} = (Fv)_1^{(a)} + (2a)(2a + 1)z^2(Fv)_0^{(a+1)}$$

which, taking into account $(Fv)_0^{(a)} = 0$ and $(Fv)_1^{(a)} = 0$, gives $(Fv)_2^{(a)} = 0$. Similarly with $n = 2$ in (26) we get

$$(Fv)_3^{(a)} = (Fv)_2^{(a)} + \frac{(2a)(2a + 1)z^2}{(3)(1)}z^2(Fv)_0^{(a+1)}$$

which gives $(Fv)_3^{(a)} = 0$. Then, by recurrence, it is easy to prove that $(Fv)_n^{(a)} = 0$, $n \in \mathbb{N}$. The same steps lead to the proof of $(Fv)_n^{(a)} = 0$. $\square$

**Remark 1.** *The explanation of the consequence is as follows: for some values of $a \in \{\cdots, -\frac{5}{2}, -2, -\frac{3}{2}, -1, -\frac{1}{2}, 0, \frac{1}{2}, 1, \frac{3}{2}, 2, \frac{5}{2}, \cdots\}$ and for some values of $n \in \mathbb{N} \cup \{0\}$ we have arrived to write a new identity between a well-defined series with infinitely many terms and a well-defined series with finitely many terms as follows*

- *for $a \in \mathbb{N} \cup \{0\}$ and $n \in \mathbb{N} \setminus \{1\}$ the series ${}_2F_1\left(a, a + \frac{1}{2}; -n + \frac{3}{2}; z^2\right)$ is well defined with infinitely many terms is equal to the series $\frac{z^{2a-2+2n}\sqrt{\pi}(a+\frac{1}{2})_{n-1} {}_2F_1(\frac{1}{2}-a, 1-n-a; \frac{1}{2}; \frac{1}{z^2})(z^2-1)^{-n+1-2a}}{\Gamma(n-\frac{1}{2})}$ which is well-defined with finitely many terms.*

- *For $a \in \mathbb{Z}_- \setminus \{0\}$ and $n \in \mathbb{N} \cup \{0\}$ the series ${}_2F_1\left(a, a + \frac{1}{2}; -n + \frac{3}{2}; z^2\right)$ is well defined with finitely many terms whereas the series $\frac{z^{2a-2+2n}\sqrt{\pi}(a+\frac{1}{2})_{n-1} {}_2F_1(\frac{1}{2}-a, 1-n-a; \frac{1}{2}; \frac{1}{z^2})(z^2-1)^{-n+1-2a}}{\Gamma(n-\frac{1}{2})}$ is well defined with finitely many terms provided that $1 - a < n$.*

- *For $a \in \{\frac{5}{2}, \frac{7}{2}, \cdots\}$ and $n \in \mathbb{N} \cup \{0\}$ the series ${}_2F_1\left(a, a + \frac{1}{2}; -n + \frac{3}{2}; z^2\right)$ is well defined with infinitely many terms is equal to the series $\frac{2\sqrt{\pi}(a)_n z^{2n+2a-3} {}_2F_1(1-a, \frac{3}{2}-n-a; \frac{3}{2}; \frac{1}{z^2})}{\Gamma(n-\frac{1}{2})(z^2-1)^{n+2a-1}}$ which is well-defined with finitely many terms. For $a \in \{\cdots, -\frac{5}{2}, -\frac{3}{2}, -\frac{1}{2}\}$ and $n \in \mathbb{N} \cup \{0\}$ the series ${}_2F_1\left(a, a + \frac{1}{2}; -n + \frac{3}{2}; z^2\right)$ is well defined with infinitely many terms whereas the*

*series* $\dfrac{2\sqrt{\pi}(a)_n z^{2n+2a-3} \; {}_2F_1\left(1-a,\frac{3}{2}-n-a;\frac{3}{2};\frac{1}{z^2}\right)}{\Gamma\left(n-\frac{1}{2}\right)(z^2-1)^{n+2a-1}}$ *is well defined with finitely many terms provided that* $\frac{3}{2} - a < n$.

One can use the following Maple instructions

```
restart; F1 := proc (n, a, z) options operator, arrow;
hypergeom([a+1/2, a], [-n+3/2], z^2) end proc; F2 := proc (n, a, z)
options operator, arrow; hypergeom([2*a, -n+1], [-2*n+2],
2*z/(1+z))/(1+z)^(2*a) end proc; u := proc (n, a, z) options
operator, arrow; 2*z^(2*a-3+2*n)*sqrt(Pi)*pochhammer(a, n)
*hypergeom([1-a, 3/2-n-a], [3/2], 1/z^2)
*(z^2-1)^(-n+1-2*a)/GAMMA(n-1/2) end proc;
v := proc (n, a, z) options operator, arrow;
-z^(2*a-2+2*n)*sqrt(Pi)*pochhammer(a+1/2, n-1)
*hypergeom([1/2-a, 1-n-a], [1/2], 1/z^2)
*(z^2-1)^(-n+1-2*a)/GAMMA(n-1/2) end proc;
factor(simplify(F1(2, -5, z)-F2(2, -5, z)-u(2, -5, z)));
0
factor(simplify(F1(2, -5/2, z)-F2(2, -5/2, z)-v(2, -5/2, z)));
0
factor(simplify(F1(n, a+1/2, z)-F1(n, a, z)+ (2*a+1)*z^2*F1(n-1,
a+1, z)/(2*n-3)));
0
factor(simplify(F2(n, a+1/2, z)-F2(n, a, z)+ (2*a+1)*z^2*F2(n-1,
a+1, z)/(2*n-3)));
0
factor(simplify(u(n, a+1/2, z)-v(n, a, z) +(2*a+1)*z^2*v(n-1, a+1,
z)/(2*n-3)));
0
factor(simplify(v(n, a+1/2, z)-u(n, a, z) +(2*a+1)*z^2*u(n-1, a+1,
z)/(2*n-3)));
0
```

## 4. Open Problem

By the same technique, we are working on other well-known quadratic transformations available in the literature. The work is under investigation and will form a part of the subsequent paper in this direction.

**Author Contributions:** M.J.A. and A.S.A.-M. contributed equally to this work. All authors have read and agreed to the published version of the manuscript.

**Funding:** This research was funded by Qassim university. The researchers would like to thank the Deanship of Scientific Research, Qassim University for funding the publication of this paper.

**Institutional Review Board Statement:** Not applicable.

**Informed Consent Statement:** Not applicable.

**Data Availability Statement:** Data is contained within the article.

**Acknowledgments:** The authors would like to thank either the editor and the referees for the time and effort to review this paper or Rathie A.K. for helpfull discussions. Their remarks and comments helped us to revise this paper and to make it clearer and more concise.

**Conflicts of Interest:** The authors declare no conflict of interest.

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
