# Peer review of "On a Resolution of Another Isolated Case of a Kummer’s Quadratic Transformation for 2F1"

_axioms, doi:10.3390/axioms12020221_

Round 1

Reviewer 1 Report (Previous Reviewer 2)

see uploaded report

Author Response

Reviewer 2 Report (New Reviewer)

The article concerns the resolution of another isolated case of a Kummer's quadratictransformation for 2F1.

The results contained in the work are original and constitute a valuable supplement to the current state of knowledge on this subject. Please check the markings of symbols and the numbering of patterns before publication.

Author Response

The referee wrote: The results contained in the work are original and constitute a valuable supplement to the current state of knowledge on this subject. Please check the markings of symbols and the numbering of patterns before publication.

Answer: Done.

This manuscript is a resubmission of an earlier submission. The following is a list of the peer review reports and author responses from that submission.

Round 1

Reviewer 1 Report

Please see the attached file!

Reviewer 2 Report

see uploaded report
